# Whole Exome Sequencing to Find Candidate Variants for the Prediction of Kidney Transplantation Efficacy

**DOI:** 10.3390/genes14061251

**Published:** 2023-06-11

**Authors:** Seyed Mohammad Kazem Aghamir, Hassan Roudgari, Hassan Heidari, Mohammad Salimi Asl, Yousef Jafari Abarghan, Venous Soleimani, Rahil Mashhadi, Fatemeh Khatami

**Affiliations:** 1Urology Research Center, Tehran University of Medical Sciences, Tehran P94V+8MF, Iran; 2Genomic Research Centre (GRC), Shahid Beheshti University of Medical Sciences (SBMU), Tehran 1416634793, Iran; 3Department of Applied Medicine, Medical School, Aberdeen University, Aberdeen AB24 3FX, UK; 4Deparment of Molecular Genetics, Faculty of Medicine, Mashhad University of Medical Sciences, Mashhad 1696700, Iran

**Keywords:** whole-exome sequencing, single nucleotide polymorphisms, kidney transplant, exonic/intronic

## Abstract

Introduction: Kidney transplantation is the optimal treatment strategy for some end-stage renal disease (ESRD); however, graft survival and the success of the transplantation depend on several elements, including the genetics of recipients. In this study, we evaluated exon loci variants based on a high-resolution Next Generation Sequencing (NGS) method. Methods: We evaluated whole-exome sequencing (WES) of transplanted kidney recipients in a prospective study. The study involved a total of 10 patients (5 without a history of rejection and 5 with). About five milliliters of blood were collected for DNA extraction, followed by whole-exome sequencing based on molecular inversion probes (MIPs). Results: Sequencing and variant filtering identified nine pathogenic variants in rejecting patients (low survival). Interestingly, in five patients with successful kidney transplantation, we found 86 SNPs in 63 genes 61 were variants of uncertain significance (VUS), 5 were likely pathogenic, and five were likely benign/benign. The only overlap between rejecting and non-rejecting patients was SNPs rs529922492 in rejecting and rs773542127 in non-rejecting patients’ MUC4 gene. Conclusions: Nine variants of rs779232502, rs3831942, rs564955632, rs529922492, rs762675930, rs569593251, rs192347509, rs548514380, and rs72648913 have roles in short graft survival.

## 1. Introduction

Kidney transplantation is the optimal treatment strategy for end-stage kidney disease; however, its complications may cause morbidity and mortality in transplant recipients [1]. Prediction of transplant rejection is essential and requires accurate, time-sensitive, and noninvasive biomarker platforms. The traditional method of monitoring allograft rejection using the elevated level of serum creatinine cannot precisely represent ongoing immunologic rejection [2]. These days, the genes that govern the acceptance or rejection of a transplant are at the center of attention. Several studies have shown that proteins, DNA, and RNA could be promising candidate biomarkers for monitoring renal transplantation rejection [3,4,5,6,7]. The putative genes that play a role in the survival of transplanted cells, tissues, or organs belong to a family called the major histocompatibility complex (MHC) [8]. The Cross Match is more important, followed by knowledge of the differences in HLA typing between the donor and the recipient. Host–donor matching is hugely related to the human leukocyte antigen (HLA) molecule. Class I HLA molecules are expressed in nucleated cells, and class II HLA molecules are expressed in antigen-presenting cells. These molecules play an essential role in kidney and bone marrow transplantation, where matching *HLA-A*, *-B*, and *-DR* loci are important [9]. There is still acute kidney transplant rejection, despite a higher chance of successful transplantation by HLA typing and matching. Therefore, it is believed that more genetic loci can be checked for higher success. This research highlighted critical exonic mutations and polymorphisms that could contribute to the destiny of kidney transplantation using next-generation sequencing (NGS).

## 2. Materials and Methods

### 2.1. Patients and Samples

This study was granted a permit by the Tehran University of Medical Sciences Ethics Committee (IR.TUMS.MEDICINE.REC.1400.149). All participants agreed to enter the study by signing a written informed consent. A total of 10 transplant patients were recruited from the urology research center, Tehran University of Medical Sciences, between 1 January 2019 and 31 December 2020. Detailed clinical and family histories of patients were provided through questionnaires. First, HLA typing is done, and then transplantation is scheduled based on the HLA matching result. HLA typing is necessary in kidney transplantation, as detection of foreign HLA by receiver T lymphocytes induces an immune response, leading to graft rejection. T lymphocyte activation starts a serial reaction of mediators that end in the immune system against the allograft. HLA laboratories currently perform serologic and molecular typing methods. After transplantation, the rejection and nonrejection genomes were sequenced separately to find possible polymorphisms with an impact on transplant rejection (Figure 1).

All five rejecting patients had mixed cellular and antibody-mediated rejection. They had chronic rejection, which occurs months after organ or tissue transplantation (rejection between 3 and 6 months considered in this study). First, we administered a triple immunosuppression scheme (tacrolimus, mycophenolate mofetil, and corticosteroids). Then, we evaluated patients for Polyomavirus-BK (BK) and cytomegalovirus (CMV) infections for additional treatment strategies.

About 5 milliliters of blood were collected at the time of recruitment. Serum creatinine and the CKD-EPI Creatinine Equation (2021) were used to estimate the GFR at the time of recruitment. DNA was harvested from blood samples using the QIAamp DNA Micro Kit (Cat# 56304).

### 2.2. Whole Exome Sequencing

Untargeted whole exome sequencing (WES) was used to study genetic variations in 10 recipients of transplanted kidneys. Samples were fragmented using a Covaris S2 ultrasonicator (Covaris, Woburn, MA, USA) and trained for sequencing on an Illumina HiSeq2500 using a custom DNA library preparation protocol established on the method described by Rohland et al. [10]. The end result read out was aligned to the human reference genome from 2013 (GRCh38/hg38) Sentieon BWA (Sentieon, Mountain View, CA, USA). The final reads and variant labeling were valued using the Sentieon DNAseq pipeline [11].

### 2.3. Analysis of Variants

All final labeled variants were annotated by ANNOVAR [12] and filtered according to variant function (nonsynonymous, stop gain/loss, splicing, frameshift insertion/deletion, and in-frame insertion/deletion variants) and minor allele frequencies (MAF; <0.1% global population MAF) in the Genome Aggregation Database (gnomAD) [13] and a population-level database of genomic variant frequencies derived from large-scale exome and genome sequencing data. The medical implications of functional variants were clarified using InterVar based on the Medical Genetics and Genomics and the Association for Molecular Pathology (ACMG-AMP) [14,15]. Finally, variants were categorized as “pathogenic” or “likely pathogenic”, and the inheritance prototype (dominant or recessive) was considered. In addition, several computational prediction algorithms, including SIFT, Polyphen2 (Polyphen2_HDIV and Polyphen2_HVAR), Mutation Taster, M-CAP, and LRT, were used to support the understanding of nonsynonymous variants categorized as “variants of unknown significance” (VUSs) according to ACMG-AMP guidelines [16]. Additional data of pathogenicity was checked for VUSs if at least 5 of these six prediction methods were in agreement with pathogenicity [16]. Furthermore, we presented copy-number variant (CNV) analysis via Atlas-CNV, a system for distinguishing and selecting high-confidence CNVs from targeted NGS data. Entirely listed CNVs were visualized using the Integrative Genomics Viewer software program and subsequently double-checked manually.

### 2.4. Statistical Analysis

Continuous data are usually shown as the median, and discontinuous data are shown as n (%). Differences between the rejecting and non-rejecting transplant groups were measured by χ2 tests, and then Fisher’s exact test (or test for trend) was used for discontinuous variables. The unpaired *t*-tests from STATA software version 17.0 were used to compare the continuous data. The SNP-set (Sequence) Kernel Association Test (SKAT) was used to assess the gene-based rare variant association burden [17]. Two-tailed *p*-values < 0.05 were considered statistically significant.

## 3. Results

Ten kidney transplantation patients from the Urology Research Center were enrolled in this study, including five rejections of transplants and five sex- and age-matched non-rejecting cases. Then, all patients were genetically sequenced for genetic alterations. Demographic information is presented in Table 1.

The median age was 37 and 36 in rejecting and non-rejecting patients. Most patients in our rejecting group were not diabetic.

Using an untargeted sequencing panel, we identified 194 rare but functional variants (including nonsynonymous, stop gain/loss, splicing, frameshift insertion/deletion, and in-frame insertion/deletion variants) in 10 patients (Appendix A). Among the 10 patients included in this study, we found 610 variants, which were already recorded in exonic, exonic splicing, intergenic, intronic, non-coding RNA exonic, splicing, untranslated upstream region, and one untranslated region.

There were no putative variants among the five transplant rejecting patients; however, there were nine variants from six genes that could be suggested to have a role in graft rejection in four patients (Table 2). All SNPs were categorized as pathogenic or likely pathogenic based on ACMG-AMP guidelines; another was classified as variants of uncertain significance “VUSs”. To distinguish between benign and pathogen variants, we used three famous sites, including Varsome (https://varsome.com/, accessed on 1 August 2022), Intervar (https://wintervar.wglab.org/, accessed on 1 August 2022), and Frankin (https://franklin.genoox.com/, accessed on 1 August 2022). Two variants were reported as likely pathogenic, including rs779232502 in *Asporin (ASPN*) and rs3831942 in *Potassium Calcium-Activated Channel Subfamily N Member 3 (KCNN3)*. Six variants were reported as likely benign, including rs564955632, rs529922492, rs762675930, rs569593251, rs192347509, and rs548514380. The only SNP reported as benign was rs72648913 on *Titin (TTN).*

In five patients with successful kidney transplantation (meaning no rejection for more than five years), interestingly, alterations in 63 known genes were found (Table 3). Out of the 86 SNPs identified in those 63 genes, 61 SNPs were VUS, five were likely pathogenic, and another five were likely benign. All variants were located on the exonic region, the coding part of a gene, but there was one variant from Acyl-CoA Synthetase Family Member 3 (ACSF3), consisting of four inserted nucleotides (GGAG) in a splicing site.

Mutation in the LVRN gene (one of those 63 genes that produces a protein called Laeverin) is associated with a disease that causes Astigmatism. This one, in fact, contributes to metallopeptidase activity, which in turn affects placentation by regulating the biological activity of the critical peptides at the embryo-maternal interface.

The *MUC4* gene (Mucin c, which is linked to pancreatic adenocarcinoma) is the principal constituent of mucus. This glycoprotein has essential functions in the protection of epithelial cells by maintaining membrane integration and has been observed to contribute to epithelial renewal and differentiation. The MUC4 gene has a coding sequence with a variable number (>100) of 48 nucleotide tandem repeats, which contributes to the O-linked glycosylation pathway, so defective C1GALT1C1 causes Tn poly agglutination syndrome (TNPS).

*TTN* (Titin) encodes a large protein for striated muscles with two ends called an N-terminal I-band and a C-terminal A-band as the elastic part of the molecule. It contains two regions of tandem immunoglobulin domains on either side of a PEVK region that is rich in proline, glutamate, valine, and lysine. The rs762675930 of TTN contains an exonic and missense replacement of Asn > Asp on amino acid 5165. Titin also includes binding sites for serving as an adhesion template to build up the contractile machinery inside muscle cells. Mutations in this gene might be associated with familial hypertrophic cardiomyopathy nine and/or the production of autoantibodies against Titin, especially in Scleroderma. The gene ontology (GO) annotations for this gene include calcium ion binding and chromatin binding. 

*Asporin* (*ASPN*) is an extracellular cartilage protein from a small leucine-rich proteoglycan family. This protein can regulate chondrogenesis by inhibiting the expression of the transforming growth factor-beta gene in cartilage.

The last one is Potassium *Calcium-Activated Channel Subfamily N Member 3* (*KCNN3*), which belongs to the *KCNN* family of potassium channels and includes two CAG repeat regions that increase susceptibility to disease. There are some spliced transcript variants that encode different isoforms of this gene.

There was no report in ClinVar for variant rs564955632 (Missense Variant), rs529922492 (Missense Variant), rs762675930 (Missense Variant), rs72648913 (Missense Variant), rs192347509 (Missense Variant), rs548514380 (Missense Variant), and rs779232502 (Missense Variant). The rs569593251 variant of the *TTN* gene is reported by ClinVar as likely benign for interpreted conditions, such as Myopathy myofibrillar with early respiratory failure. rs3831942 is an in-frame deletion, which plays a role in Pyloric stenosis and Esophageal atresia.

These pathogenic or likely pathogenic single nucleotide and insertion/deletion variants were detected in 3 genes, including *ZNF806*, *HYDIN*, and *ATXN3*. There were another five benign or likely benign single-nucleotide and insertion/deletion variants detected in 5 genes, too, including *FAM104B*, of *MUC12*, *ARMC9*, *ZNF717*, and *MUC4*, the only gene shared between both rejecting and the non-rejecting groups was *MUC4*; however, the variants were different, including rs529922492 in the rejecting group and rs773542127 in the non-rejecting group. The medications used for patients were the same between rejecting and non-rejecting patients regardless of the variants (Table 4).

## 4. Discussion

Using next-generation sequencing (NGS) technology, we successfully identified nine variants that could predict kidney transplantation rejection and 86 variants that might play a protective role. Genetic testing for managing chronic kidney disease and transplantation is rapidly developing. Available guidelines from the OPTN (Organ Procurement and Transplantation Network) and KDIGO (KDIGO Clinical Practice Guideline on the Evaluation and Management of Candidates for Kidney Transplantation) imply that genetic testing in a renal transplant should become more widespread [18,19,20,21]. We found nine polymorphisms from 6 genes as promoting and 86 protective variants for rejection of kidney transplants. These six genes and their related proteins are shown in Figure 2.

These genes co-occur with the biological term “kidney transplant” in the GeneRIF Biological Term Annotations dataset. They are from different genes, including Adiponectin, C1Q, and collagen domain-containing (ADIPOQ) and cytochrome P450, family 3, subfamily A, polypeptide 5 (CYP3A5). (https://maayanlab.cloud/Harmonizome/gene_set/kidneytransplant/GeneRIF+Biological+Term+Annotations, accessed on 1 August 2022).

It has been recently published by Mann et al. that nearly one-third of pediatric renal transplant recipients had a genetic cause of their kidney disease identified by WES [22]. For the first time, Mota-Zamorano and colleagues found that some variants in the *LEPR* and *ADIPOQ* genes of donors and recipients might affect the outcome of kidney transplantation [23]. Later, it was observed in several studies that *ADIPOQ* played an essential role in patients with post-transplant diabetes mellitus [24,25,26,27,28].

We detected pathogenic variants in three genes, including *ZNF806*, *HYDIN*, and *ATXN3.* The only variant that we found in *KCNN3* was the insertion of fifteen nucleotides (GCTGCTGCTGCTGCT) that led to a frameshift mutation resulting in a nonfunctional protein. *KCNN3* plays a role in different pathways, so changes in its activity can affect some physiological mechanisms. Calcium-activated K+ via its specific channels promotes action potential (AP) repolarization; accordingly, some *KCNN2* and *KCNN3* variants are associated with increased atrial fibrillation (AF) risk. In addition, histone deacetylase-related epigenetic mechanisms have been found to affect AP regulation [29]. Thus, epigenetic change of *KCNN3* was reported in new-onset diabetes after kidney transplant (NODAT), which had an adverse impact on kidney allograft and patient survival [30]. Myeloid-derived suppressor cells (MDSC) form a heterogeneous population of immature myeloid cells that increasingly proliferate in inflammatory conditions, including transplantations, and KCNN3 is top listed among the 50 genes involved in the migration of MDSCs [31]. A large-scale study also identified a relationship between *KCNN3* and *IL6R* genes and AP [32].

*LVRN* (Laeverin) encodes a cell surface aminopeptidase that is involved in embryonic signaling pathways. LVRN belongs to the M1 peptidase family, also called ‘aminopeptidase Q’ [33,34]. Primate *LVRN* has a unique peptide-binding motif (HXMEN), where the first glycine (Gly) residue is substituted by histidine (His), inducing significant changes in end-product peptide hormones. However, there are no previous studies on the role of *LVRN* in kidney transplantation; for the very first time, we suggest that *LVRN* plays an essential role in the rejection of kidney transplants.

The *TTN* we studied is rs762675930, an exonic and missense alteration affecting amino acid 5165 (Asn > Asp). Previously, *TTN* was also evaluated for its role in cardiac arrhythmias, cardiomyopathies, and sudden death [35].

The *MUC4* gene is the only gene shared between rejecting and non-rejecting groups. However, the detected variants were different (rs529922492 in rejecting and rs773542127 in non-rejecting). There are eight well-known human epithelial mucin genes [36] that several studies have implied have roles in transplantation. In 2003, Wasserberg and colleagues suggested that early transplant rejection could be associated with increased MUC2, MUC4, IFN-gamma, and TNF-alpha expression [37], so they can be used as predictive markers in combination with histopathologic examination for assessment of the risk of graft rejection. A study by Shamloo and colleagues on MHC Class I and II mismatch using a mouse model suggested a role for mucins in the pathogenesis of dry eye-associated with graft versus host disease. MUC4 has also been reported as an ERK signaling pathway activator in epithelial cells. An animal study suggested a role for MUC4 and ERK signaling pathways in oxidative stress and CaOx crystal formation in renal tubular epithelial cell [38].

The rs192347509 of *SVEP1* was identified in two Short segment Hirschsprung disease patients [39]. The *SVEP1* gene encodes a large extracellular matrix protein with sushi (complement control protein), von Willebrand factor type A, epidermal growth factor-like (EGF), and pentraxin domains (PMID: 11062057, PMID: 16206243), so its deficiency leads to increased plasma levels of Cxcl1, which is an expansion of plaque inflammatory leukocytes that in turn promotes atherosclerotic plaque formation. Coxam and colleagues reported that *SVEP1* plays the role of a modulator for vessel anastomosis during developmental angiogenesis in zebrafish embryos, so the loss of SVEP1 followed by a decrease in blood flow together contributes to a defective anastomosis of intersegmental vessels [40]. Accordingly, we assume that the inhibition of the degradation of *SVEP1* gene products using products may offer a therapeutic strategy to prevent kidney transplant rejection [41].

*ASPN* encodes a protein that belongs to a family of leucine-rich repeat (LRR) proteins that exist in the cartilage matrix. The name asporin shows the exceptional aspartate-rich N terminus and its connection to decorin [42]. Studies on ASPN have found a promotional role for *ASPN* in epithelial and mesenchymal transformation and in invasion, migration, and metastasis of several malignancies via activating the CD44-AKT/ERK-NF-kappaB pathway [43,44,45]. Several studies have found the periodontal ligament-associated protein-1 (*PLAP-1*)/*asporin*’s susceptibility gene for osteoarthritis. *PLAP-1*/*asporin* negatively regulates TLR2- and TLR4-induced inflammatory responses through direct molecular interactions [46]. Iida and coworkers found a SNP that maps within a 33.4-kb genomic region covering *ASPN* [47]. Sakashita and colleagues showed that the absence of *PLAP-1* could inhibit high-fat diet-induced metabolic syndrome and bone resorption in vivo, and adipocyte differentiation resulting in an extracellular matrix change. Studying *PLAP-1* represents the correlation between diabetes and periodontal disease [48]. Mice lacking *PLAP-1*/*asporin* responds to extraordinary fat diet-induced metabolic disorder and alveolar bone loss by using adipose tissue expansion [48].

The presence of the C allele of rs1044250 and the G allele of rs2278236 in the *ANGPTL4* gene is linked to the developed risk of moderate/severe proteinuria in renal transplant patients [49], but this was not the case in our study. We found pathogenic single nucleotide and insertion/deletion variants in *ZNF806*, *HYDIN*, and *ATXN3* genes among non-rejecting patients; however, none of the previous studies could see these three genes’ role in transplant rejection. Some investigators suggested a higher incidence of *ZNF 469* gene variants in fast progressive advanced keratoconus patients who had surgery by the age of 30 compared to its frequency in the average Turkish population [50]. *ZNF* genes have been shown to act as potential molecular biomarkers involved in RCC carcinogenesis. The limitations of this research include the very small number of patients who were studied.

## 5. Conclusions

It is quite applicable to use the WES test to predict the survival of kidney transplantation. Moreover, nine variants rs779232502, rs3831942, rs564955632, rs529922492, rs762675930, rs569593251, rs192347509, rs548514380, and rs72648913 of six genes, LVRN, MUC4, TTN, SVEP1, ASPN, and KCNN3, have possible roles in graft rejection that are suggested to be checked in graft recipients.

## Figures and Tables

**Figure 1 genes-14-01251-f001:**
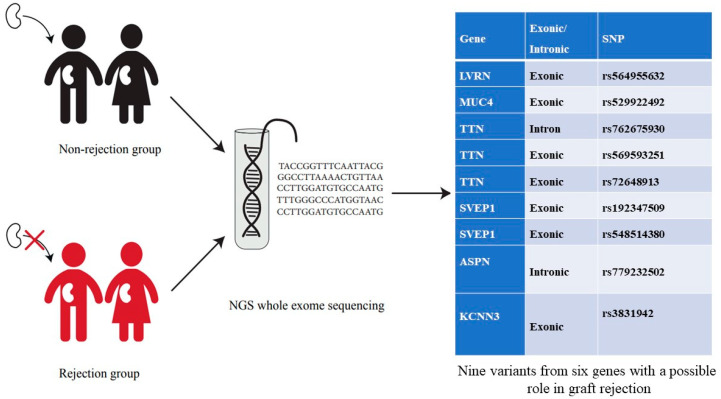
Graphical abstract of the research procedure.

**Figure 2 genes-14-01251-f002:**
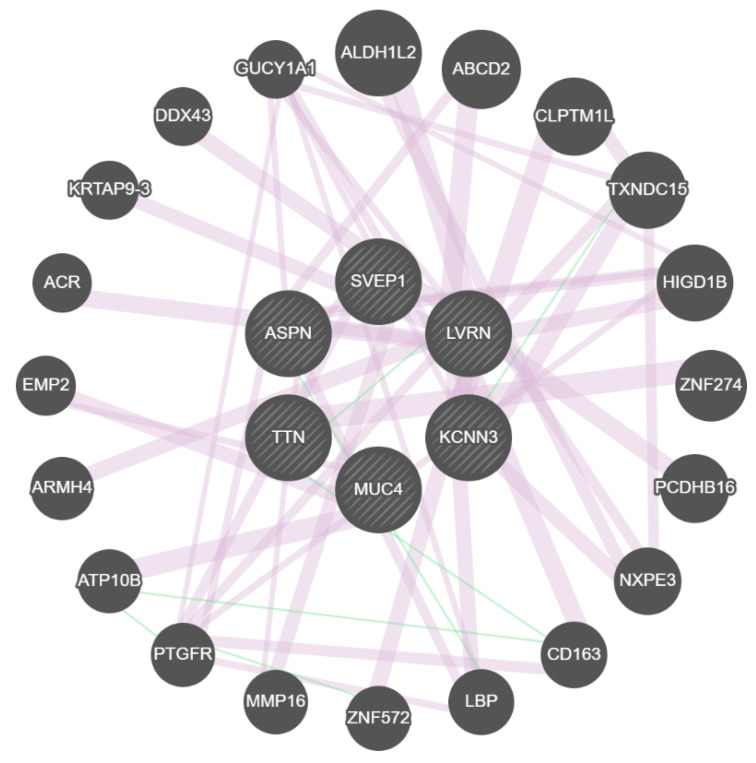
The protein interaction network of six target genes with other genes based on GeneMANIA.

**Table 1 genes-14-01251-t001:** Demographic information of the study subjects.

Variable	Group		*p*-Value
	Rejecting	Non-Rejecting	
Sex (male), n (%)	4 (80%)	4 (80%)	1.000
Education (BSc), n (%)	1 (20%)	3 (60%)	0.524
Age (year), median (IQR)	37 (36–43)	36 (34–42)	0.459
Creatinine, median (IQR)	5 (3.6–6)	1.3 (1.2–1.3)	0.009
Weight, median (IQR)	76 (70–76)	65 (65–87)	0.750
Height, median (IQR)	170 (168–170)	168 (168–168)	0.142
BMI, median (IQR)	24.8 (24.0–26.3)	23.9 (23.0–30.4)	0.917
DM (yes), n (%)	2 (40%)	1 (20%)	0.500
DM family history (yes), n (%)	2 (40%)	1 (20%)	0.500

**Table 2 genes-14-01251-t002:** The characteristics of 9 variants from six known genes with possible role in graft rejection.

	Gene	Exonic/ Intronic	SNP	Pathogenicity	Cytoband	Start	End	REF Sequence	ALT Sequence	Depth
1	*LVRN*	Exonic	rs564955632	VUS	5q23.1	115320380	115320380	G	A	78
2	*MUC4*	Exonic	rs529922492	VUS	3q29	195510751	195510751	G	T	455
3	*TTN*	Intron	rs762675930	VUS	2q31.2	179611634	179611634	C	T	199
4	*TTN*	Exonic	rs569593251	VUS	2q31.2	179406087	179406087	G	A	105
5	*TTN*	Exonic	rs72648913	Benign	2q31.2	179610967	179610967	C	T	166
6	*SVEP1*	Exonic	rs192347509	Likely Benign	9q31.3	113170636	113170636	C	T	98
7	*SVEP1*	Exonic	rs548514380	VUS	9q31.3	113261413	113261413	C	T	146
8	*ASPN*	Intronic	rs779232502	VUS	9q22.31	95236968	95236968	A	G	33
9	*KCNN3*	Exonic	rs3831942	Likely Benign	1q21.3	154842199	154842199	-	GCTGCTGCTGCTGCT (insertion)	80

**Table 3 genes-14-01251-t003:** The characteristics of 90 variants of 63 common genes with a possible role in preventing graft rejection.

	Gene	Exonic/Intronic	SNP	Pathogenicity	Cytoband	Start	End	REF Sequence	ALT Sequence	Depth
1	*ACSF3*	Splicing	-	-	16q24.3	89167068	89167068	-	GGAG	44
2	*ANKRD36*	Exonic	rs71329611	VUS	2q11.2	97864333	97864334	CC	AT	36
3	*ANKRD36C*	Exonic	rs78179792	VUS	2q11.1	96616504	96616504	C	A	103
4	*ANKRD36C*	Exonic	rs78178577	VUS	2q11.1	96521640	96521640	C	T	211
5	*ANKRD36C*	Exonic	rs76276218	VUS	2q11.1	96517474	96517474	G	A	130
6	*ANKRD36C*	Exonic	rs200183690	VUS	2q11.1	96521090	96521090	T	C	108
7	*ANKRD36C*	Exonic	rs77124870	VUS	2q11.1	96517471	96517471	A	G	128
8	*ATXN3*	Exonic;splicing	rs748879218	Likely Pathogenic	14q32.12	92537397	92537397	T	-	73
9	*CDC27*	Exonic	rs111227623	VUS	17q21.32	45216162	45216162	A	C	94
10	*CDC27*	Exonic	rs76836152	VUS	17q21.32	45216172	45216172	A	G	94
11	*CDC27*	Exonic	rs62075657	VUS	17q21.32	45216132	45216132	T	G	93
12	*CDC27*	Exonic	rs796538886	VUS	17q21.32	45219329	45219329	A	C	87
13	*CTDSP2*	Exonic	rs74343811	VUS	12q14.1	58220841	58220841	C	T	168
14	*CTDSP2*	Exonic	rs111346934	VUS	12q14.1	58220823	58220823	C	T	165
15	*CTDSP2*	Exonic	rs76940645	VUS	12q14.1	58220816	58220816	A	G	165
16	*CTDSP2*	Exonic	rs75591888	VUS	12q14.1	58220844	58220844	C	T	168
17	*CTDSP2*	Exonic	rs74554628	VUS	12q14.1	58240210	58240210	G	T	115
18	*DSPP*	Exonic	-	-	4q22.1	88537133	88537133	G	A	38
19	*FAM104B*	Exonic	rs1047037	Likely Benign	Xp11.21	55172687	55172687	T	C	213
20	*FRG1BP; FRG1DP*	Exonic	rs867116961	VUS	Xp11.21	29612099	29612099	G	A	151
21	*FRG1DP; FRG1BP*	Exonic	rs1047037	VUS	Xp11.21	29612103	29612103	T	C	203
22	*FRG2C*	Exonic	rs2118760	VUS	3p12.3	75715099	75715099	T	A	235
23	*HLA-A*	Exonic	-	-	6p22.1	29911260	29911261	AC	CG	116
24	*HLA-B*	Exonic	-	-	6p21.33	31324887	31324888	GG	CT	73
25	*HLA-DQA1*	Exonic	rs386699859	VUS	6p21.32	33037639	33037640	GC	AT	38
26	*HLA-DQB1*	Exonic	-	-	6p21.32	32632659	32632660	CT	TG	57
27	*HLA-DRB1*	Exonic	rs796196270	VUS	6p21.32	32551957	32551958	GC	TT	82
28	*HLA-DRB1*	Exonic	rs796846373	VUS	6p21.32	32551938	32551939	GG	AT	79
29	*HYDIN*	Exonic	rs375727122	-	16q22.2	70954704	70954718	GCGCTCCTTCTCCGT	-	182
30	*HYDIN*	Exonic	rs11337008	Likely Pathogenic	16q22.2	70896016	70896016	A	-	111
31	*IGSF3*	Exonic	rs75947003	VUS	1p13.1	117142700	117142700	C	A	107
32	*IGSF3*	Exonic	rs61786651	VUS	1p13.1	117156459	117156459	C	T	117
33	*KRTAP9-1*	Exonic	rs76389571	VUS	17q21.2	39346578	39346578	C	G	68
34	*MUC2*	Exonic	-	-	11p15.5	1093483	1093484	GT	AC	302
35	*MUC3A*	Exonic	rs113251740	VUS	2q21.2	100551133	100551133	C	T	513
36	*MUC3A*	Exonic	rs72494466	VUS	7q22.1	100551122	100551122	T	G	448
37	*MUC3A*	Exonic	rs760637480	VUS	7q22.1	100551094	100551094	G	A	423
38	*MUC3A*	Exonic	rs771974573	VUS	7q22.1	100551092	100551092	G	C	158
39	*MUC3A*	Exonic	rs749410668	VUS	7q22.1	100551082	100551082	C	T	170
40	*MUC3A*	Exonic	rs141925032	VUS	7q22.1	100550704	100550704	C	A	139
41	*MUC4*	Exonic	rs773542127	Likely Benign	3q29	195510254	195510254	C	G	160
42	*MUC6*	Exonic	rs374545453	VUS	11p15.5	1017239	1017240	CG	TA	263
43	*OR2L8*	Exonic	rs34851853	VUS	1q44	248112762	248112763	TG	CA	78
44	*PCM1*	Exonic	rs754721723	VUS	8p22	17796382	17796383	AC	GT	69
45	*PLIN4*	Exonic	-	-	19p13.3	4513143	4513144	CA	TG	107
46	*PSG3*	Exonic	rs34721205	VUS	19q13.2	43243217	43243218	AA	GG	85
47	*TEX11*	Exonic	rs386825673	VUS	Xq13.1	69749852	69749853	AT	GA	43
48	*ZNF717*	Exonic	rs796081257	VUS	3p12.3	rs796081257	rs796081257	CA	GG	144
49	*PRB3*	Exonic	-	-	12p13.2	11420495	11420496	GG	AA	53
50	*RP1L1*	Exonic	rs747592079	VUS	8p23.1	10466686	10466686	G	A	103
51	*SLC35G4*	Exonic	rs386801281	VUS	18p11.21	11609903	11609904	CA	GG	150
52	*ZNF705E*	Exonic	-	-	1q13.4	71527895	71527896	CG	TA	162
53	*ZNF717*	Exonic	rs77322475	Likely Benign	3p12.3	75787159	75787159	C	T	325
54	*ZNF806*	Exonic	rs11491243	VUS	2q21.2	133075612	133075612	C	T	181
55	*ZNF806*	Exonic	rs113311843	Likely Pathogenic	2q21.2	133075904	133075904	-	A	154
56	*ZNF806*	Exonic	rs111405036	Likely Pathogenic	2q21.2	133076118	133076118	A	-	146
57	*ZNF806*	Exonic	rs111944984	Likely Pathogenic	2q21.2	133075479	133075479	C	-	221
58	*ARHGEF26*	Exonic	rs386667246	VUS	3q25.2	153839959	153839960	CT	TC	87
59	*ARMC9*	Exonic	rs386656198	Likely Benign	2q37.1	232087474	232087475	AT	GA	66
60	*MYG1, MYG1-AS1, PFDN5*	Exonic	rs71453838	VUS	12q13.13	53693532	53693533	AA	GC	43
61	*DUSP5*	Exonic	rs35834951	VUS	10q25.2	112266822	112266823	GC	AT	62
62	*FCGBP*	Exonic	rs796880559	VUS	19q13.2	40368498	40368499	AA	CC	71
63	*GTF2IRD2; GTF2IRD2B*	Exonic	rs370642824	VUS	7q11.23	74558397	74558397	C	A	24
64	*HLA-C*	Exonic	rs796075923	VUS	6p21.33	31238009	31238010	TT	CG	31
65	*IP6K3*	Exonic	rs34332988	VUS	6p21.33	33690796	33690797	CA	TG	87
66	*KLRC3*	Exonic	rs796361824	VUS	12p13.2	10573094	10573095	CA	GG	16
67	*KRTAP10-6*	Exonic	-	-	21q22.3	46012181	46012182	CG	TA	49
68	*MPP2*	Exonic	rs70964679	VUS	17q21.31	41960633	41960634	CG	GC	83
69	*MUC12*	Exonic	rs763405288	Likely Benign	7q22.1	100639418	100639419	CG	AA	50
70	*OR2L8*	Exonic	rs34851853	VUS	1q44	248112762	248112763	TG	CA	102
71	*OR2T35*	Exonic	-	-	1q44	248801610	248801611	GC	AT	35
	*OR9G1; OR9G9*	Exonic	rs71458233	VUS	11q12.1	56468047	56468048	AC	GT	278
72	*PCM1*	Exonic	rs754721723	VUS	8p22	17796382	17796383	AC	GT	71
73	*PSG3*	Exonic	rs34721205	VUS	19q13.2	43243217	43243218	AA	GG	98
74	*RGPD5*	Exonic	-	-	2q13	110593536	110593536	C	A	14
75	*RNF212*	Exonic	-	-	4p16.3	1087329	1087329	-	TGGAGCCAGCCAT	44
76	*SCARF2*	Exonic	rs70944210	VUS	22q11.21	20779946	20779947	CG	GC	37
77	*SPIB*	Exonic	rs113934432	VUS	19q13.33	50926264	50926265	TG	CC	40
78	*TCF15*	Exonic	rs71212728	VUS	20p13	590542	590543	CG	GC	55
79	*TRIM50*	Exonic	rs71517080	VUS	7q11.23	72738762	72738763	CA	TG	65
80	*VCX2*	Exonic			Xp22.31	8138170	8138171	CT	GC	19
81	*PCDHA9*	Exonic	rs35021536	VUS	5q31.3	140230370	140230371	AA	CC	115
82	*MUC5B*	Exonic	-	-	11p15.5	1258240	1258241	CA	TG	158
83	*OR4C3*	Exonic	rs386753295	VUS	11p11.2	48346961	48346962	AA	TG	174
84	*KRTAP12-2*	Exonic	rs35163632	VUS	21q22.3	46086757	46086758	GC	AT	83
85	*PIGZ*	Exonic	rs71611508	VUS	3q29	196674972	196674973	CT	TC	69
86	*PRIM2*	Exonic	-	-	6p11.2	57512796	57512796	-	CA	86

VUS: variants of uncertain significance.

**Table 4 genes-14-01251-t004:** Detailed information on medications in five rejecting (columns 1 to 5) and five non-rejecting patients (columns 6 to 10).

	Patient Code									
	1	2	3	4	5	6	7	8	9	10
Mediation	B1 100	Vit B6	Rocatrol	Rocatrol	Vit B6	Rocatrol	Metoral	Metoral	Rocatrol	Nistatine
	Metoral	Amilodip	Amilodip	Amilodip	Prazosin	Calcium D	A.S.A	A.S.A	Calcium D	A.S.A
	Amilodip	Ranitiine	NEFROVIT	NEFROVIT	Nistatine	VALCYTE	Vit B6	Vit B6	Carvedilol	Vit B6
	Nistatine	Metoral	Allopurinol	Metoral	Rocatrol	Vit B6	Pantamine	Pantamine	Cinacalcet	Carvedilol
	Rocatrol	Nistatine	Tacrolimus	Nistatine	Amilodip	Amilodip	Pantaprozol	Pantaprozol	Diltiazem	Ranitiine
	Sevelamer	Pantaprozol	Losartan	Pantaprozol	Calcium D	Nistatine	CellCept	CellCept	Losartan	Rocatrol
	VALCYTE	VALCYTE	Omeprazole	VALCYTE	VALCYTE	Allopurinol	Clotrimazole	Clotrimazole	Metformin	VALCYTE
	Levofloxacin	Insulin	Atorvastatin	Insulin	Pantaprozol	Isoniazid	Folic Acid	Folic Acid	Pantamine	Furosemide
	Pantaprozol	Allopurinol	CellCept	Allopurinol	Atorvastatin	Pantaprozol	Prednisolone	Prednisolone	Pantaprozol	Metformin
	Clotrimazole	CellCept	Co-trimoxazole	CellCept	CellCept	Atorvastatin	Rocatrol	Rocatrol	Amlopidine	Pantaprozol
	NEFROVIT	Clotrimazole	Folic Acid	Clotrimazole	Clotrimazole	CellCept			Persulfate	Tamsulosin
	Prednisolone	Folic Acid	Hydrochlorothiazide	Folic Acid	Folic Acid	Clotrimazole			NEFROVIT	CellCept
	Sandimmune	Prednisolone	Prednisolone	Prednisolone	Prednisolone	Folic Acid			Omeprazole	Clotrimazole
		Rocaltrol	Sandimmune	Rocaltrol		Prednisolone			Atorvastatin	Folic Acid
		Sandimmune		Sandimmune		Sandimmune			CellCept	Prednisolone
									Clotrimazole	
									Prednisolone	

## Data Availability

All necessary data are submitted through manuscript. Detailed data will be available on request.

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
