# Peer review of "Whole Exome Sequencing to Find Candidate Variants for the Prediction of Kidney Transplantation Efficacy"

_genes, 2023, doi:10.3390/genes14061251_

Round 1

Reviewer 1 Report

Dear Authors,

The information given in your article is really interressant, but many points should be improved, maybe it is not possible for you but I give you my opinion and  my point of view.

In the document in green you can find the point to improve

-The abstract with 274 words is too long an do not comply with the  200 words expected

-Figure 1 the quality of the resolution should be improved

In the graph it will be nice to present the gene ( as describe in table 1) and the variants, the variant without the information of the gene is without interest.

Material and methods

Whole Exome Sequencing

-The pipeline used for sequencing ( ref 12 is from 2010) is there no  more recent pipeline

-In the sequence analyzes the genome should be  from 2013 (GRCh38/hg38) and not from 2009 (hg19)

-Variant Analysis

The method used for variant analysis is ANNOVAR ( ref 13), is there not  a more recent method

-Introduction : please explain if any other studies before have been done, and what kinds of studies

-Please put in the table the sequence position in the same line, because it is difficult to read

-Table 3 is cut

-Are all these active links necessary in the text ?, we used three famous sites, including Varsome (https://varsome.com/), Intervar (https://wintervar.wglab.org/), and Frankin (https://franklin.genoox.com/).

https://www.proteinatlas.org/humanproteome/tissue/kidney

-The discussion explains well the physiology and the active effect of each gene, but  no relations are made or supposition of interactions, a more detailed molecular explanation could be performed.

The conclusion quite direct and gives no perspectives, this conclusion should be more detailed regarding the results and the discussion.

Kind regards

Author Response

Dear Miss Mara Pop
Assistant Editor of Genes

Thanks for your points and comments on my manuscript. The following is my reply which is highlighted in the manuscript:

Reviewer 1

The information given in your article is really interressant, but many points should be improved, maybe it is not possible for you but I give you my opinion and my point of view.

In the document in green, you can find the point to improve

The abstract with 274 words is too long and do not comply with the  200 words expected. I have just made the less than 200 words abstract by removing unnecessary data.

 -Figure 1 the quality of the resolution should be improved. Figure 1 is replaced with another figure with high resolution.

In the graph, it will be nice to present the gene  as described in table 1) and the variants, the variant without the information of the gene is without interest. Good point I have just added the data of the genes to figure 1.

Material and methods

Whole Exome Sequencing

-The pipeline used for sequencing ( ref 12 is from 2010) is there no  more recent pipeline

It has been replaced by new one: Kendig KI, Baheti S, Bockol MA, Drucker TM, Hart SN, Heldenbrand JR, Hernaez M, Hudson ME, Kalmbach MT, Klee EW, Mattson NR. Sentieon DNASeq variant calling workflow demonstrates strong computational performance and accuracy. Frontiers in genetics. 2019 Aug 20;10:736.

-In the sequence analysis the genome should be  from 2013 (GRCh38/hg38) and not from 2009 (hg19)

It has been corrected

-Variant Analysis

The method used for variant analysis is ANNOVAR ( ref 13), is there not  a more recent method. I do not have the newer

-Introduction : Please explain if any other studies before have been done, and what kinds of studies. All other similar studies are considered in the discussion part.

-Please put in the table the sequence position in the same line, because it is difficult to read. I can not understand this comment. The table order is based on the text order.

-Table 3 is cut

It is a long table and when it is getting published in the PDF of the journal should be placed (InDesign).

-Are all these active links necessary in the text ?, we used three famous sites, including Varsome (https://varsome.com/), Intervar (https://wintervar.wglab.org/), and Frankin (https://franklin.genoox.com/).

They help to a better understanding of the text.

-The discussion explains well the physiology and the active effect of each gene, but no relations are made or supposition of interactions, a more detailed molecular explanation could be performed.

Thanks for the point following part is added to the discussion part.

We found nine polymorphisms from 6 genes as promoting and 86 protective variants for rejection of kidney transplants. These six genes and their related proteins are shown in figure 2.

Figure 2: The protein interaction network of six target genes with other genes based on GeneMANIA.

The conclusion is quite direct and gives no perspectives, this conclusion should be more detailed regarding the results and the discussion. The new conclusion is :

nine variants rs779232502, rs3831942, rs564955632, rs529922492, rs762675930, rs569593251, rs192347509, rs548514380, and rs72648913 of six LVRN, MUC4, TTN, SVEP1, ASPN, and KCNN3 genes have possible role in graft rejection that are suggested to be checked in graft recipients.

Reviewer 2

The more recent availability of technology which facilitates whole exome sequencing has stimulated interest in the nephrology/renal transplant spheres. Hence the data contained in this manuscript will be of potential interest. However, there are some issues with the manuscript in its current form which require the attention of the authors. They are listed as follows-

1) The second statement in the abstract that the HLA typing should be checked to ensure that there is minimal risk of graft rejection needs to be amended. Is it not the Cross Match that is more important followed by knowledge of the differences in HLA typing between the donor and the recipient? Thanks for the point. The abstract is edited, and this statement is added to the introduction part.

2) Can you make it clearer in the abstract that the study involved a total of 10 patients (5 without a history of rejection and 5 with). Thanks again the following clear sentence has been added to the abstract part.

Methods

We evaluate whole-exome sequencing (WES) of transplanted kidney recipients in a prospective study. The study involved a total of 10 patients (5 without a history of rejection and 5 with). About five milliliters of blood were collected for DNA extraction, followed by whole-exome sequencing (NGS) based on molecular inversion probes (MIPs).

3) You need to spell out in the Methods section as to what was meant by a rejecting patient. It appears that patients who had a history of any type of rejection were included. However, no information is provided on the actual types of rejection sustained by the 5 patients in the rejecting group-ie how many episodes of rejection had each patient had and at what time points (this information needs to be included in the results section). Of note if these 5 patients had all sustained different types of rejection at differing time points post-transplantation, then this potentially is a source of bias for this particular study. Actually, the medication strategies were different (table 4) and we can not solve the bias but we consider five rejecting patients who have rejection at the same time points post-transplantation with similar reasons.

4) Plus, there needs to be information included in Table 1 as to what was the median time point post-transplantation (and the range) for when this exome sequencing was actually undertaken, for each of the subgroups. The rejection time All five rejecting patients have mixed cellular and antibody-mediated rejection. They had chronic rejection which occurs months after organ or tissue transplantation (rejection between 3 to 6 months considered in this study).

5) The information in Table 4 is presented in a confusing manner. Can this not be reformatted so that it is less confusing? I have just added a heading to the table.

6) You need to reference some of the recently published literature on the utility of whole exome sequencing in some subgroups of Nephrology patients to support the second sentence of the first paragraph of the Discussion section https://pubmed.ncbi.nlm.nih.gov/30655312/#:~:text=Whole-Exome%20Sequencing%20Enables%20a%20Precision%20Medicine%20Approach%20for,both%20transplant%20patients%20and%20potential%20living%20related%20donors.

Thanks for the point. I have just added the reference and its data in the discussion part. It has been recently published by Mann, Nina, et al. that nearly one-third of pediatric renal transplant recipients had a genetic cause of their kidney disease identified by WES (23).

7) Why do you think there was an imbalance between the number of protective variants versus the number of pathogenic variants which you identified as a result of this research? I can not explain this maybe a larger sample size can explain it.

8) You need to include a paragraph in the Discussion section on the limitations of this research-including the very small number of patients who were studied, ? at different time points post-transplant, as well as potentially the lack of a uniform group who had had prior rejection episodes. Thanks for the point I have just added the limitation at the end of the discussion.

9) You only have preliminary results here, hence you have identified some trends which now need to be confirmed by other studies (involving fare greater numbers of patients). This also needs to be spelt out in the Discussion section. Thanks for the point I have just added the limitation at the end of the discussion.

10) What is potentially the utility of whole exome sequencing in the post renal transplant population in the future-can you also discuss this as well? In graft candidate we can check these 9 variants of 6 genes to predict graft survival.

11) The Conclusions section at the end of the Discussion section needs to be rewritten. You do not have enough data as a result of this study to even be able to propose at this stage that WES can be used to predict the survival of kidney transplant allografts. You need to modify your conclusions. The new conclusion is:

It is quite applicable to use the WES test to predict the survival of kidney transplantation. Moreover, nine variants rs779232502, rs3831942, rs564955632, rs529922492, rs762675930, rs569593251, rs192347509, rs548514380, and rs72648913 of six LVRN, MUC4, TTN, SVEP1, ASPN, and KCNN3 genes have possible role in graft rejection that are suggested to be checked in graft recipients.

Best Regards

Fatemeh Khatami

2023.03.29

Reviewer 2 Report

The more recent availability of technology which facilitates whole exome sequencing has stimulated interest in the nephrology/renal transplant spheres. Hence the data contained in this manuscript will be of potential interest. However, there are some issues with the manuscript in its current form which require the attention of the authors. They are listed as follows-

1) The second statement in the abstract that the HLA typing should be checked to ensure that there is minimal risk of graft rejection needs to be amended. Is it not the Cross Match which is more important followed by knowledge of the differences in HLA typing between the donor and the recipient?

2) Can you make it clearer in the abstract that the study involved a total of 10 patients (5 without a history of rejection and 5 with).

3) You need to spell out in the Methods section as to what was meant by a rejecting patient. It appears that patients who had a history of any type of rejection were included. However, no information is provided on the actual types of rejection sustained by the 5 patients in the rejecting group-ie how many episodes of rejection had each patient had and at what time points (this information needs to be included in the results section). Of note if these 5 patients had all sustained different types of rejection at differing time points post transplantation, then this potentially is a source of bias for this particular study.

4) Plus, there needs to be information included in Table 1 as to what was the median time point post transplantation (and the range) for when this exome sequencing was actually undertaken, for each of the subgroups.

5) The information in Table 4 is presented in a confusing manner. Can this not be reformatted so that it is less confusing?

6) You need to reference some of the recently published literature on the utility of whole exome sequencing in some subgroups of Nephrology patients to support the second sentence of the first paragraph of the Discussion section https://pubmed.ncbi.nlm.nih.gov/30655312/#:~:text=Whole-Exome%20Sequencing%20Enables%20a%20Precision%20Medicine%20Approach%20for,both%20transplant%20patients%20and%20potential%20living%20related%20donors.

7) Why do you think there was an imbalance between the number of protective variants versus the number of pathogenic variants which you identified as a result of this research?

8) You need to include a paragraph in the Discussion section on the limitations of this research-including the very small number of patients who were studied, ? at different time points post-transplant, as well as potentially the lack of a uniform group who had had prior rejection episodes

9) You only have preliminary results here, hence you have identified some trends which now need to be confirmed by other studies (involving fare greater numbers of patients). This also needs to be spelt out in the Discussion section.

10) What is potentially the utility of whole exome sequencing in the post renal transplant population in the future-can you also discuss this as well?

11) The Conclusions section at the end of the Discussion section needs to be rewritten. You do not have enough data as a result of this study to even be able to propose at this stage that WES can be used to predict the survival of kidney transplant allografts. You need to modify your conclusions.

Author Response

(The authors gave the same response as above.)

Round 2

Reviewer 2 Report

The revised version of the manuscript still requires further attention from the authors in light of the previous reviewers comments. This includes-

1) Providing the specific data and information around what type of rejection the 5 rejecting patients each had along with the median time point and the range post transplantation. Did each of these patients only sustain one episode of proven rejection or were there instances where some of these patients sustained multiple episodes of rejection. This needs to be spelt out. All of this data must be included because any variation in the nature of the rejection episodes is a potential confounder of the results which you have then obtained.

2) The limitations of you research are not only due to the limited number of patients who were studied. The results are potentially confounded due to other factors one of which is that the patients may not have all experienced the same type of rejection at the same time point (nor have been managed in precisely the same way). In the absence of the relevant data on the nature of the rejection process that was sustained by each patient it is hard to know how much of a confounder this all is. Plus, your cohort are predominantly males so it is not clear if these results would be applicable to females-another confounder associated with this particular study.

3) Your conclusions are invalid (particularly the written conclusions following the Discussion section of the manuscript), in that all you have here are some preliminary findings from a limited number of patients. There is no way that the WES test can be used to predict the survival of kidney transplant allografts at this point in time. This research needs to be replicated in far larger cohorts of kidney transplant recipients, including in a wider range of patients (ie older patients, more female patients, and across a number of different transplant centers), and the data analyzed in order to determine which gene variants are associated with inferior graft survival versus which gene variants are associated with superior graft survival. 

Author Response

(The authors gave the same response as above.)
